# *Leishmania infantum* Infection of Primary Human Myeloid Cells

**DOI:** 10.3390/microorganisms10061243

**Published:** 2022-06-17

**Authors:** Morgane Picard, Calaiselvy Soundaramourty, Ricardo Silvestre, Jérôme Estaquier, Sónia André

**Affiliations:** 1Faculty of Science, Université Paris Cité, INSERM U1124, F-75006 Paris, France; picard.morgane.noelie@gmail.com (M.P.); selvys93@yahoo.fr (C.S.); smc.andre88@gmail.com (S.A.); 2Life and Health Sciences Research Institute (ICVS), School of Medicine, University of Minho, 4710-057 Braga, Portugal; ricardosilvestre@med.uminho.pt; 3ICVS/3B’s, PT Government Associate Laboratory, 4710-057 Braga/Guimarães, Portugal; 4CHU de Québec, Université Laval Research Center, Faculty of Medecine, Québec City, QC G1V 4G2, Canada

**Keywords:** *Leishmania*, flow cytometry, blood, monocyte, neutrophil

## Abstract

Circulating phagocytic cells often serve as cellular targets for a large number of pathogens such as *Leishmania* parasites. Studying primary human cells in an infectious context requires lengthy procedures for cell isolation that may affect the analysis performed. Using whole blood and a no-lyse and no-wash flow cytometric assay (*NoNo* assay), we monitored the *Leishmania infantum* infection of primary human cells. We demonstrated, using fluorescent parasites, that among monocyte cell populations, *L. infantum* preferentially infects classical (CD14^+^CD16^−^) and intermediate (CD14^+^CD16^+^) primary human monocytes in whole blood. Because classical monocytes are the preponderant population, they represent the larger *L. infantum* reservoir. Moreover, we also found that, concomitantly to monocyte infection, a subset of PMNs is infected early in whole blood. Of interest, in whole blood, PMNs are less infected compared to classical monocytes. Overall, by using this *NoNo* assay, we provided a novel avenue in our understanding of host–leishmania interactions.

## 1. Introduction

*Leishmania donovani* and *infantum* are intracellular parasites belonging to the *Trypanosomatidae* family that are the causative agents of visceral leishmaniasis (VL) [1]. This disease affects vulnerable populations, and approximately 90,000 infections occur each year [1,2]. The clinical manifestations of VL are fever, anemia, weight loss, splenomegaly and hepatomegaly, leading to death if left untreated [3]. *Leishmania* parasites are transmitted as promastigotes (extracellular forms) to the mammalian host during phlebotomine sandfly blood meals. Thus, parasites are introduced into the skin following sand fly inoculation [4,5]. In turn, tissue damage due to the inoculation of parasites causes the recruitment and accumulation of immune cells at the bite’s site, involving monocytes and polymorphonuclear neutrophils (PMNs). In addition, it has been reported that sand fly salivary proteins attract and activate host cells, and taking advantage of this inflammatory response, parasites invade their host [6]. PMNs may act as a Trojan horse, silently transmitting parasites to infiltrate mononuclear cells [4,5,7,8,9,10]. Furthermore, *Leishmania* hijack host machinery to survive and multiply into host cells [11,12]. Of importance, the parasite has also been detected in the peripheral blood of patients with cutaneous or visceral leishmaniasis [13,14]. Risks of the contamination of leishmaniasis via blood transfusion or organ donation (bone marrow and liver) in endemic regions where asymptomatic cases predominate are not negligible [14,15,16]. However, little attention has been paid regarding the interaction between viscerotropic strains and monocyte subsets.

Human monocytes are heterogeneous populations including three main circulating subsets that are defined in humans as classical (CL: HLA-DR^+^CD14^+^CD16^−^), intermediate (ITM: HLA-DR^+^CD14^+^CD16^+^) and nonclassical (NCL: HLA-DR^+^CD14^low^CD16^+^) [17,18,19]. CL monocytes are attracted by CCL2 and CCL3 gradients and are more efficient than ITM monocytes in producing ROS and pro-inflammatory molecules, which highlights their potential to migrate to injured or inflamed tissues [20]. On the other hand, NCL populations possess a more mature phenotype and are associated with a gene expression profile related cell-to-cell adhesion, cell trafficking, proliferation, and differentiation. Thus, they express higher levels of CX3CR1, which contribute to migration and adherence to fractalkine-secreting endothelium [21]. ITM and NCL populations bridge the gap between innate and adaptive immunity. Thus, among their many functions, they are both involved in T-cell proliferation and stimulation [18,19]. Furthermore, in infectious and inflammatory conditions, these two populations are amplified [22,23,24]. They have redundant functions with classical monocytes; in fact, ITM populations are able to produce ROS, and NCL populations produce pro-inflammatory cytokines such as TNF-α and IL-1β [25]. Previous reports on *L. braziliensis*, responsible for cutaneous leishmaniasis, have demonstrated that promastigotes infect CL and ITM monocytes [26], while amastigotes do not preferentially target a monocyte subset but rather infect the three subsets [20]. Thus, studying the dynamics of parasite interaction with myeloid cells in the blood may provide information about the dissemination of viscerotropic strains from the periphery to the organs. 

Regarding PMNs, it is now becoming clear that they are also heterogeneous cells with potentially multiple subsets in health and disease [27,28]. PMNs are the most abundant type of circulating myeloid cells constituting around 50% of peripheral blood, being identified in flow cytometry by CD16 positivity and HLA-DR negativity. Recent studies indicated an in vitro polarization of PMNs toward N1- and N2-like phenotypes, mirroring macrophage polarization, in which the latter is less potent in killing parasites than that observed in the presence of N1 [27]. However, PMNs are short-lived cells once isolated, and most of the studies directed at understanding parasite infection use PMNs isolated after density gradient centrifugation [29,30,31,32,33,34,35]. However, consideration should be given to cell isolation that can modulate the cell surface expression of markers such as CCR2 and CD14 [36,37,38,39,40]. 

Thereby, by using a no-lyse and no-wash assay, using flow cytometry, we assessed the *L. infantum* infection of monocytes and PMNs concomitantly in whole blood. This assay gave us an extensive vision of infected cells without the need for cell isolation and consequently no cell marker loss. Thus, we demonstrated that *L. infantum* infects the different subsets of myeloid cells with a selective tropism for CL and ITM populations, with PMNs being less infected as compared to the former. Interestingly, a subset of PMNs seemed to be preferentially infected. This assay provided a novel avenue to analyze host–pathogen interaction, which limits cell manipulation, in greater depth.

## 2. Methods

### 2.1. CFSE Labeling of Leishmania infantum

Metacyclic promastigotes of *L. infantum* (*Li*, MHOM/MA/67/ITMAP-263) were cultured under axenic culture (at 26 °C), and low passage was used to guarantee infectivity. Parasites were labeled with carboxyfluorescein diacetate succinimidyl ester (CFSE, ThermoFischer Scientific, Montigny le Bretonneux, France), as described by Moreira et al. [41]. Briefly, promastigotes were counted using a Kova slide and resuspended in 2 mL of PBS containing 5 µM of CFSE for 1.2 × 10^8^ parasites and incubated for 10 min at 37 °C and then washed. The labeled promastigotes suspension was divided. One part was heat-inactivated at 60 °C for 15 min (heat-killed, HK) and validated using microscopy (non-motile parasites). The two suspensions of parasites (live and HK) were used for infections.

### 2.2. Whole-Blood Infection

Whole blood from healthy donors (between 18 and 50 years old) was provided by Etablissement Français du Sang (R75069, CPD-SAGM whole blood), and all donors provided informed consent to EFS. In order to monitor the dynamic of interaction between the cells and parasites, the infection rate was assessed at two times (4 and 24 h). Whole blood (1 mL), transferred in a 15 mL tube (Falcon tube, BD Biosciences, Le Pont-de-Claix, France), was infected with either live parasites at three doses (1 × 10^6^, 5 × 10^6^ and 10 × 10^6^ of parasites per mL) or HK parasites (10 × 10^6^ parasites per mL). For each condition, whole blood was incubated at 37 °C and 5% CO_2_ before analysis.

### 2.3. Flow Cytometry

Whole blood was stained with LIVE/DEAD fixable violet stain (ThermoFisher Scientific, Montigny le Bretonneux, France), CD3-PerCP (clone SP34-2, BD Biosciences, Le Pont-de-Claix, France), CD20-PE (clone 2H7, BD Biosciences, Le Pont-de-Claix, France), HLA-DR-ECD (clone immu357, Beckman Coulter, Villepinte, France), CD8-PECy7 (clone RPA-T8, BD Biosciences, France), CD14-APC (clone TÜK4, Miltenyi Biotec, Paris, France) and CD16-APC-AF750 (clone 3G8, Beckman Coulter, Villepinte, France). Monocytes and PMNs were analyzed in whole blood after fixation with paraformaldehyde (1%). Cells were analyzed using an Attune NxT acoustic focusing cytometer (ThermoFischer Scientific, Montigny le Bretonneux, France) with a violet scatter filter (NxT No-Wash No-Lyse filter) installed with a collection rate of 200 µL/min. Analyses of two time points (4 and 24 h) were performed at the same time in order to attain comparative analyses of fluorescence intensities.

### 2.4. Confocal Microscopy

Leukocytes were isolated after cell lysis with NaCl 0.2% and submitted to magnetic separation with CD14 Microbeads (130-050-201, Miltenyi Biotec, Paris, France). The cells’ cytoplasms were stained with CellTracker^TM^ Deep Red (C34565, Invitrogen, Villebon-sur-Yvette, France) for 45 min at 37 °C. After being washed several times and fixation, cells were visualized with a Zeiss LSM 710 confocal microscope.

### 2.5. Statistical Analyses

All values were expressed as means ± SEM. Statistical analysis was performed using the non-parametric paired Wilconxon test or the Two-Way ANOVA test with a Tukey analysis and a Spearman test for correlation. Comparisons were considered significant when *p* < 0.05. All statistical analyses were performed with GraphPad Prism 8 software (San Diego, CA, USA).

## 3. Results

### 3.1. L. infantum Infects Monocytes in Whole Blood

The method that we used to monitor infection exploited the difference between red blood cells that contain hemoglobin, which absorbs violet laser (405 nm) light, and leukocytes, which do not. Thus, by coupling 405 nm (violet) and 488 nm (blue) laser excitations to study whole-blood cells, a unique side scatter (SSC) pattern was observed (Figure 1A(a)), separating erythrocytes from leukocytes and platelets [42]. Cell populations were then discriminated using a cocktail of specific antibodies. The gating strategy used to identify the myeloid populations is shown in Figure 1. After selecting white cells from whole blood, leukocytes were defined on SSC and FSC parameters (Figure 1A(b)), and the analysis was performed on single cells (Figure 1A(c)) and excluded dead cells using the LIVE/DEAD probe. Then, T lymphocytes (CD3^+^) were excluded (Figure 1A(d)), as well as the CD3^−^CD8^+^ (natural killer cells) and CD20^+^ (B cells) populations (Figure 1A(e)). To discriminate myeloid cell populations, we used HLA-DR expression, because it was only expressed on monocytes. Thus, by combining HLA-DR and CD16 markers, (Figure 1A(f)), we separated PMNs (CD16^+^DR^−^) from monocytes (DR^+^) independently to be CD16^−^ or CD16^+^.

To distinguish between the infection versus uptake of parasites via myeloid cells, *L. infantum* was killed at 60 °C. Firstly, we validated the CFSE staining of labeled live and dead (heat-killed) parasites showing similar fluorescence intensities after in vitro treatment for both live (geometric mean fluorescence intensity (geo MFI), 5782) and HK (geo MFI, 5938) parasites (Figure 1B). Then, we assessed the percentages of infected monocytes (DR^+^) via flow cytometry after 4 h of infection with an increasing dose of parasites in whole blood (Figure 1C(a)). Hence, with 1 × 10^6^ parasites per mL, 11.4 ± 2.4% of monocytes were infected (Figure 1C(a–b)), whereas with higher doses (5 × 10^6^ or 10 × 10^6^ parasites), the percentages of *L. infantum*-infected monocytes were relatively similar (31.4 ± 5.0% and 39.8 ± 5.0%, respectively). The infection of monocytes increased with the time of parasite infection (Figure 1C(b)). Thus, the percentages of infected cells at 24 h increased to 22.2 ± 8.0% with 1 × 10^6^ parasites. With 5 × 10^6^ and 10 × 10^6^ parasites, these rates reached a plateau of 50.6 ± 9.3% and 52.5 ± 8.5%, respectively (Figure 1C(b)). Of note, the percentages of HK-labeled monocytes were 30.1 ± 3.7% at 4 h and 39.9 ± 10.3% at 24 h, which were lower compared to live parasites, particularly early after infection. Thus, *L. infantum* promastigotes can infect monocytes in whole blood without cell isolation.

### 3.2. L. infantum Preferentially Infects CL and ITM Monocytes in Whole Blood

To further characterize the nature of infected monocyte cell subsets, we analyzed the three main subsets, CL, ITM and NCL (Figure 2A). Furthermore, the double-negative population (DN: HLA-DR^+^CD14^−^CD16^−^) was also analyzed (Figure 2A). Firstly, we analyzed the distribution of the parasite among monocyte cell subsets (Figure 2B). According to the intensity of *L. infantum*-CFSE^+^ in each quadrant of the heat-map view (Figure 2B), 4 h after infection, most of the *L. infantum* parasites were localized in the CL monocytic region (88.4 ± 1.8% at the lowest dose), whereas less than 5% of the parasites were observed in the other regions. Increasing the doses of parasites, we did not observe major changes in the distribution of parasites between the different subsets (Figure 2C). HK parasites were also mostly observed in the region of CL (78.3 ± 5.5%) compared to in ITM (3.5 ± 0.8%), NCL (8.1 ± 2.4%) and DN subsets (10.0 ± 4.4%) (Figure 2C). To validate the concept that parasites infect cells in whole blood and do not remain at the surface, we purified monocytes using anti-CD14 beads which included both CL and ITM. The fraction of cells was analyzed using confocal microscopic analysis. The images showed that both live and dead *L. infantum* parasites were detected inside the cells (Figure 2D). Interestingly, the images showed the transition of the living parasites into amastigotes (1–5 µm), while, as expected, the HK parasites remained in the promastigote form (10–15 µm).

Because CL monocytes are the preponderant population in whole blood, using flow cytometry, we assessed the frequency of infected cells in each subset at two time points (4 and 24 h, Figure 3). It is interesting to note that at higher doses (5 × 10^6^ and 10 × 10^6^ parasites), the rates of infection were quite similar at both time points (Figure 3A,B), reaching 60.8 ± 10% of CL infection with 10 × 10^6^ parasites at 4 h and 74.6 ± 10.5% at 24 h (Figure 3B), whereas this frequency of infected CL was 15.2 ± 3.7% with 1 × 10^6^ parasites at 4 h and tended to be higher at 24 h (26.6 ± 10% *p*-value = 0.1484). 

Interestingly, whereas the ITM population is not the predominant population in whole blood, we found that the frequency of *L. infantum*-infected ITM populations was elevated at 4 h (51.0 ± 9.3% of ITM populations were infected with 10 × 10^6^ parasites) and quite similar to that observed for CL populations (60.8 ± 10%). After 24 h, the rate of infection increased with a maximum reached with 10 × 10^6^ parasites (74.6 ± 10.5% for CL and 65.0 ± 12.3% for ITM populations). Of importance, at lower doses, this increase was particularly significantly higher after 24 h compared to 4 h (29.4 ± 9.8% versus 12.1 ± 2.3%, *p*-value = 0.0391) of infection, reaching percentages closer to those observed in CL populations (Figure 3A,B). Of note, while the same number of parasites was used for HK and live leishmania to infect whole blood, the frequencies HK parasites’ uptake by the cells were lower at 4 h (CL, 49.8 ± 10.4% and ITM, 35.6 ± 9.3%) to those observed with live parasites in CL (60.8 ± 10%) and ITM (51.0 ± 9.3%) subsets (CL, *p*-value = 0.0391 and ITM, *p*-value = 0.1875).

In contrast to that observed for CL and ITM populations, early after infection (4 h) the frequencies of *L. infantum*-infected NCL and DN populations were low (Figure 3A,B), even at the higher dose of infection (15.6 ± 2.6% and 6.1 ± 1.6%, respectively). After 24 h, the rates of infection increased for these two subsets with a maximum reached with 10 × 10^6^ parasites (25.5 ± 9.0% for NCL, and 21.1 ± 6.6% for DN). We also analyzed the geo MFI of infected CL populations at 4 and 24 h, showing quite similar intensities, except for HK, in which the geo MFI was higher at 24 h compared to live parasites (Figure 3C).

Altogether, our results demonstrated that CL and ITM populations are preferentially infected cell subsets in whole blood despite the fact that the latter is less abundant compared to CL populations.

### 3.3. Infection by L. infantum Impacts Monocyte Cell Subsets in Whole Blood

Because *L. infantum* could impact the nature of infected monocyte cell subsets, we analyzed their percentages in whole blood. As mentioned, CL was the most abundant monocyte subset (73.9 ± 3.6%) in the whole blood of healthy donors, whereas ITM and NCL subsets represented 3.2 ± 0.4% and 8.0 ± 0.4%, respectively (Figure 4). The DN subsets represented 15.5 ± 4.1% of blood monocytes (Figure 4A). With 10 × 10^6^ parasites, CL represented 64.5 ± 6.3% in the presence of live parasites and 54.1 ± 9.0% for HK in comparison to 73.9 ± 3.6% in non-infected cells. We did not observe such significant differences for ITM and NCL populations (Figure 4A). Thus, both live and HK parasites induced the lowest CL levels in whole blood. This decrease in the percentage of CL was associated with an increase in the percentage of DN, which may be suggestive of a CD14 down-regulation in CL (Figure 4(Bb)). The absence of a significant difference between live parasites was related to the fact that a decrease was only observed in four donors out of the six. Thus, we found that the percentage of CL was negatively correlated with the rate of CL infection 4 h post-infection (Figure 4A,(Ba)). Thus, the lowest infection rate was associated with the highest percentage of CL. Therefore, the increase rate of DN infection that we observed in whole blood may represent infected, CL in which CD14 is lost. 

Altogether, these results clarify the impact of *L. infantum* infection on monocyte cell subsets in whole blood. 

### 3.4. L. infantum Infects PMNs in Whole Blood

Because this assay allowed us to monitor concomitantly with the monocyte PMN population, we assessed the rate of PMN infection in the same whole blood samples. Early after infection, we detected PMN infection (Figure 5A). The percentages of CFSE^+^-PMN reached 10 ± 4.3% at the dose of 1 × 10^6^ (Figure 5B). Little impact on the infection rate was observed with higher doses, where 5 × 10^6^ and 10 × 10^6^ parasites led to 14.4 ± 5.0% and 17.0 ± 5.1% of infected PMNs, respectively. These percentages were lower compared to the extent of infection of CL (60.8 ± 10% at the same time using 10 × 10^6^ parasites). Interestingly, PMNs were capable to phagocytize HK parasites (25.37 ± 10.9% of PMNs at 4 h) (Figure 5B).

Our results also indicated that the rate of infection was quite similar between 4 and 24 h with 1 × 10^6^ parasites (9.2 ± 5.5%), while these proportions declined using higher doses (10 × 10^6^, 8.8 ± 2.8% compared to 17.0 ± 5.1% (*p*-value = 0.0234) (Figure 5A,B). The geo MFI indicated the lower intensity of infected PMNs with live parasites at 24 h compared to 4 h as well as for HK parasites (Figure 5C), and the intensity differed to that observed in the same conditions for CL geo MFI (Figure 3C).

As shown here, among PMNs, we identified two populations, in which one expressed a higher level of CD16 expression (CD16^bright^). By plotting CD14 against CD16, this population was represented in whole blood (49.5 ± 9.9% ranging from 21 to 77.3% of PMNs in the non-infected condition) (Figure 5D). Thus, we defined PMNs as CD14^low^CD16^+^ and CD14^bright^CD16^bright^ cell subsets. We observed that the percentages of CD14^bright^CD16^bright^ PMNs were lower in the presence of increased doses of parasites compared to the non-infected condition (Figure 5D-E). Thus, more than 40% of this population disappeared 4 h after infection and even more drastically after 24 h at the highest doses (5 × 10^6^ and 10 × 10^6^ parasites). Analyzing the tropism of *L. infantum*-infected PMNs in more detail, we observed that this CD14^bright^CD16^bright^ population displayed most of the CFSE-labeled *L. infantum* parasites early after infection, as shown by the heat-map view (Figure 5D). Thus, the diminution of these PMN subsets at the highest dose may support the observation of the lowest infection rate observed at 24 h (Figure 5B). When increasing the number of parasites, most CFSE-labeled *L. infantum* parasites were then detected in the CD14^low^CD16^+^ population (Figure 5D). Of interest, our results highlighted that at lower doses of parasites (1 × 10^6^ parasites per mL), this PMN population persisted even after 24 h of infection in whole blood. 

Collectively, our results highlighted that PMNs can be infected in whole blood, although the percentages of infected cells were lower as compared to CL monocytes. We further demonstrated the selective tropism of *L. infantum* infection for PMN subsets, in which the majority of the parasites at lower doses infected and persisted in CD14^bright^CD16^bright^, which may represent a “sanctuary” for *L. infantum*.

## 4. Discussion

This flow cytometric assay demonstrated the efficacy of monitoring human myeloid cells’ infection via visceral *Leishmania* species directly on whole blood. This designed strategy allowed us to accurately study *L. infantum* target cells, demonstrating that promastigotes infect human whole-blood cells, in which CL and ITM populations are the preferential targets along with a preferential subset of PMNs. These results represent, to our knowledge, the first demonstration of human monocytic tropism via visceral parasite strains in whole blood.

Whereas numerous studies have described macrophage infection, less attention has been paid regarding human monocytic cell subsets. Classical monocytes represent 80–95% of circulating human monocytes, whereas ITM and NCL populations represent 2–5% and 2–10%, respectively [17,18,19]. Our results demonstrated that from whole blood, *L. infantum* infects CL and ITM populations preferentially. Thus, more than 50% of cell subsets were infected at the highest dose of parasites. This preference was also highlighted by the fact that even at lower doses, these two populations were targeted by parasites compared to NCL or DN cell subsets. However, given the proportion of CL populations in the blood, this population represents a major target for parasites. This propensity of infection differs from previous studies using a mucocutaneous *Leishmania* species. Indeed, the authors observed that more than 85% of the cells were infected with *L. braziliensis* amastigotes without any specific preference among the different monocytic cell subsets [20]. In contrast, Polari et al. [26]. reported that the infection of PBMCs via *L. braziliensis* promastigotes leads to a preferential infection of CL and ITM populations compared to NCL populations. A difference in the rate of infection could be related to the use of amastigote vs. promastigote parasites. Herein, *L. infantum* promastigotes also preferentially infected CL and ITM populations in whole blood. CL populations have a wide range of functions, including a great capacity for phagocytosis, ROS production and immune sensing [17], while ITM populations participate in Ag-presentation and subsequently in T-cell stimulation [18,19]. Furthermore, CL and ITM monocytes were shown to be better in phagocytizing GFP-expressing *E. coli* [43] in comparison to NCL monocytes. In our study, similarly to live parasites, we found that HK parasites were engulfed by CL and ITM populations better than NCL populations. Novais et al. [20] proposed that CL monocytes control *L. braziliensis* infection by ROS production, while ITM populations promote pathogenesis [20,43]. In the case of viscerotropic strains, it was revealed that monocytes of patients with VL had a decrease in ROS production [44]. Given the ability of CL and ITM populations to migrate into inflamed or damaged tissues, this preferential infection may provide support to parasite dissemination within tissues. In a study performed in rodents, the authors showed that pro-inflammatory monocytes (Ly6C^Hi^), besides being a preferential target for *L. donovani*, were associated with a pathogenic role [45], promoting dissemination in the liver and spleen. Whether the infection of ITM populations contributes to favor parasite visceralization in humans remains an open question.

Concomitantly to monocyte infection, we also reported the infection of PMNs. Whereas other studies, using purified PMNs, reported that *L. infantum* can infect 60% of PMNs [44,46], our results indicated lower levels (less than 25%). In the setting of the experiment performed, using whole blood and not purified PMNs, we cannot exclude the possibility that those lower levels of PMN infection reflect competition between the different target cells. Moreover, our results also highlighted the preferential targeting of CD14^bright^CD16^brigh^ PMNs. Thus, this approach by using whole blood revealed interesting findings. Firstly, the observation of this PMN subset can be explained by the absence of density gradient centrifugation, which modulates cell surface expression [40]. Secondly, this specific PMN population was lost by increasing the quantity of parasites and time of exposure, suggesting the short life of cells once infected at higher doses of parasites. This could be consistent with CD16 lowering, a marker of activated and apoptotic PMNs [47]. The role of *Leishmania* in modulating apoptosis as a mechanism employed by the parasites to manipulate PMNs and guarantee the establishment of a perennial infection [48] seems to be more complex. Several reports have proposed that PMNs are more prone to die when infected with *L. braziliensis* [49], *L. infantum* or *L. major* [50], whereas others have proposed that *L. major* mediates parasite survival [51]. Thus, human PMN activation and function are dependent on the parasite strain used [11,52,53]. Whereas at higher doses, this population, CD14^bright^CD16^brigh^, disappeared, at lower quantities (1.10^6^ parasites), this population persisted and was infected. Recent advances suggest, like for monocytes, the existence of distinct PMN subtypes including N1 and N2-like phenotypes, in which the N2 subset is unable to kill parasites [27]. The observation of parasite persistence in this CD14^bright^CD16^brigh^ at lower quantities may be indicative that this population represents the N2 population. Further experiments are needed to characterize this PMN population more in depth.

By using this approach to infect whole blood, our results also indicated that *L. infantum* infection impacts the proportion of CL monocytes in culture. Whereas it is not clearly defined whether *L. infantum* parasites bind to the membrane CD14 of myeloid cells, a lower level of CL populations was positively correlated with the rate of infection. This lower level may reflect CD14 shedding, which is a hallmark of activated monocytes [54,55]. Indeed, CD14 is a glycosylphosphatidyl-anchored membrane protein that can be a direct ligand for LPS and act as a co-receptor for TLR4 [54,55,56]. A classic explanation for CD14 shedding is that it could be a way for cells to protect themselves from activation. Thus, sCD14 could be in competition for PAMPs in binding limiting inflammation [55,57]. It was reported in some studies that increasing levels of sCD14 was correlated with severity [11,58] and was associated with the outcome of VL [54]. In this sense, parasites could, by an unknown mechanism, promote the release of sCD14 and thus decrease the inflammatory response by inhibiting macrophage activation [59,60]. Our results regarding lower levels of CD14 were different to the report from Melo et al. [54], which may reflect the culture conditions, since, instead of using purified macrophages, we used whole-blood cells, which include PMNs. Indeed, the human leukocyte elastase, a serine proteinase produced by PMNs, is directly involved in the cleavage of CD14 [61,62]. Thus, by using whole blood as a model of *Leishmania* infection, this may have reflected the in vivo situation, in which higher levels of sCD14 were reported in VL patients [58] or co-infected with HIV [63]. It is interesting to note that a decrease in the marker Ly6C has also been also observed in murine inflammatory monocytes infected with *L. major*, supporting the hypothesis that such negative modulation may provide a strategy for the parasite to colonize its host [9]. Thus, by analyzing the infection of whole-blood cells and by limiting cell manipulation and isolation, we provided novel insights about viscerotropic strains infecting primary human myeloid cell subsets. This flow-cytometric approach may provide a novel assay for the monitoring of *Leishmania* infection.

## Figures and Tables

**Figure 1 microorganisms-10-01243-f001:**
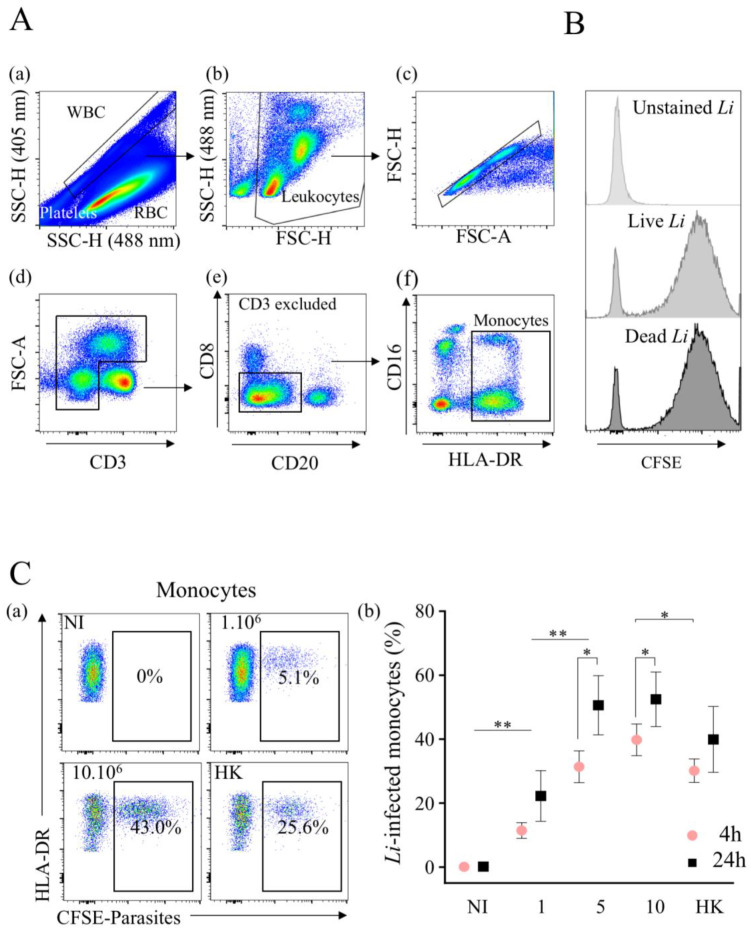
*Leishmania infantum* infects primary human monocytes in whole blood. (**A**) Immunophenotyping of whole blood from healthy donors was conducted using a No-Wash No-lyse flow cytometric approach. Here, it was the representative gating strategy. Platelets and red blood cells were excluded by combining side scatter plot (SSC) of 405 nm and 488 nm lasers (**a**). Leukocytes were visualized on FSC against SSC plot (**b**), and singlet cells (**c**) were selected for further gating. (**d**) Lymphocytes expressing CD3 were excluded as well (**e**), CD8 (NK) and CD20 (B cells) populations; (**f**) on CD8^neg^CD20^neg^ population, we analyzed the expression of CD16 against HLA-DR (monocytes). (**B**) CFSE intensity of live and dead parasites in myeloid cells. For comparison, unstained parasite is shown. (**C**) (**a**) Flow cytometry of CFSE-labeled *Leishmania infantum* in non-infected (NI) and infected monocytes with different quantities (1 × 10^6^, 5 × 10^6^ and 10 × 10^6^/mL noted 1, 5 and 10, respectively) of live or heat-killed (HK, 10 × 10^6^/mL) parasites. (**b**) The percentages of infection at 4 and 24 h are shown. Data are the mean of 7 donors. Statistical analysis was performed using one-tailed Wilcoxon test (* *p* < 0.05, ** *p* < 0.01, *n* = 7).

**Figure 2 microorganisms-10-01243-f002:**
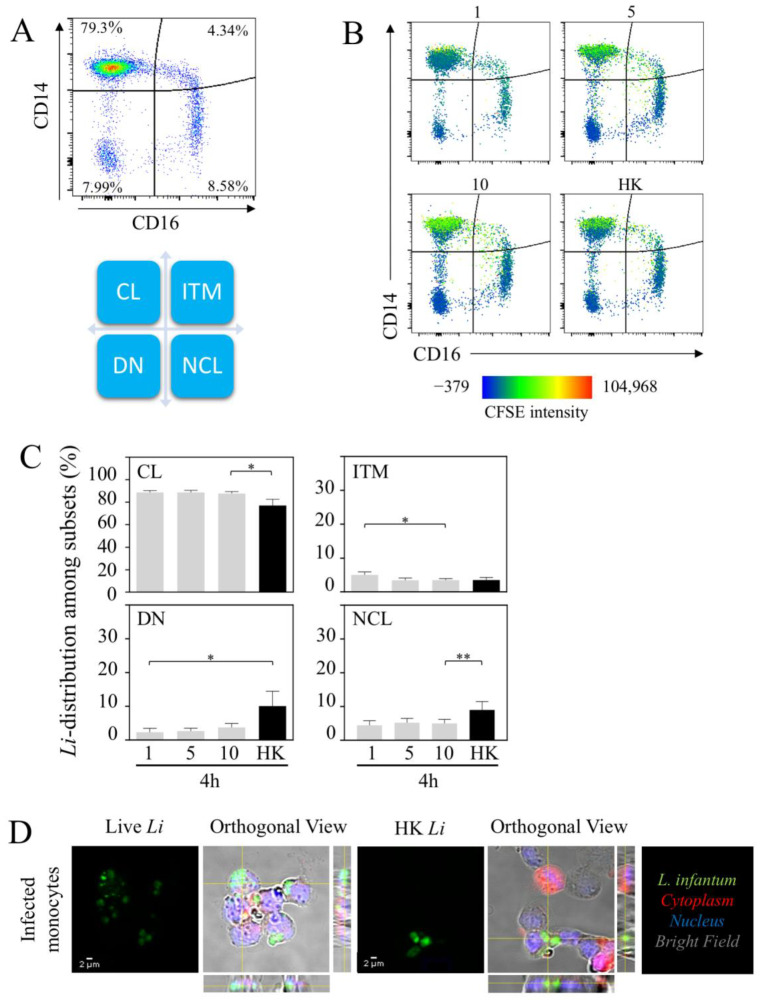
*Leishmania infantum* identifies CL as a main reservoir. (**A**) Dot plot showing the expression of CD14 and CD16 on HLA-DR+ monocytes identifying the classical (CL), the intermediate (ITM), the non-classical (NCL) and the double-negative (DN) populations. (**B**) Representative heat-map view of CFSE-labeled parasites. Color intensity of parasite is shown in each population of monocytes. (**C**) Parasite distribution in each monocyte subset 4 h after infection with 1.10^6^, 5.10^6^ or 10.10^6^ of live parasites is noted as 1, 5 and 10, respectively, and heat-killed (HK, 10.10^6^). (**D**) Confocal microscopy of monocytes infected with live or HK *L. infantum* (*Li*) in CFSE (green). Bright field associated with CellTracker allowed to determine the membrane and the cytoplasm (red) of each cell. Standard deviation intensity projection of x, y and z axes confirmed that the parasites were localized inside the cells (orthogonal view). Statistical analysis was performed using one-tailed Wilcoxon test (* *p* < 0.05, ** *p* < 0.01; *n* = 7).

**Figure 3 microorganisms-10-01243-f003:**
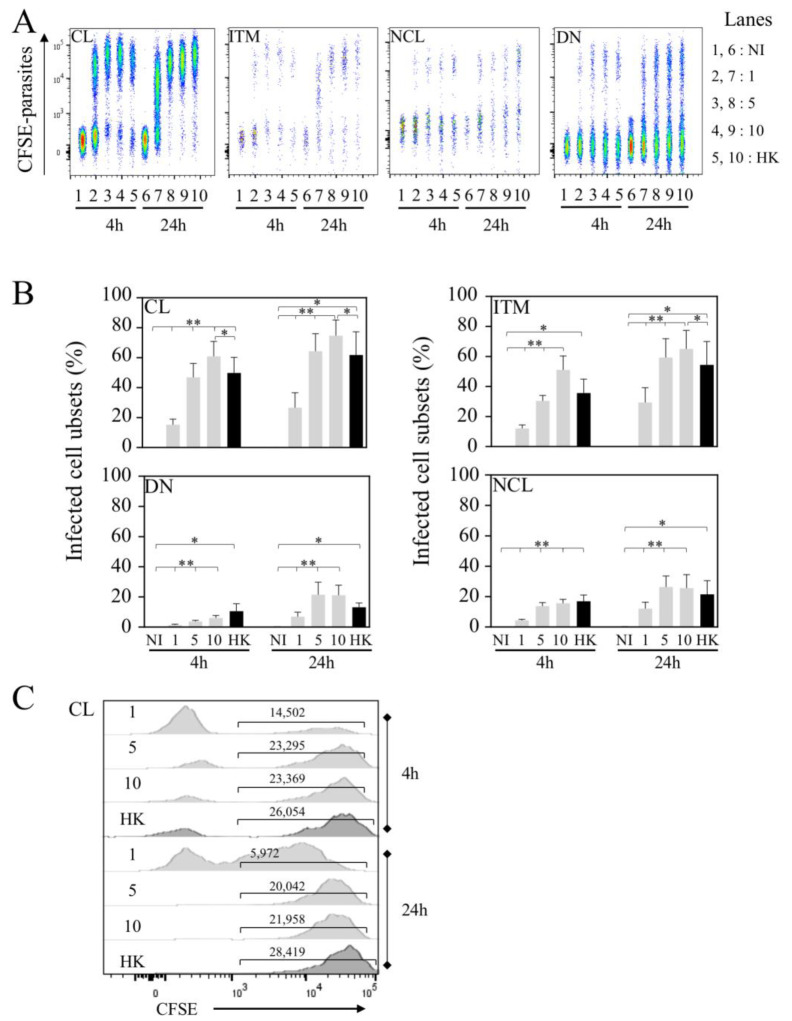
CL and ITM populations are preferentially infected by *Leishmania infantum*. (**A**) Flow cytometry of *Li*-infected cells. Representative concatenation of CFSE parasites in classical (CL), intermediate (ITM), non-classical (NCL) and double-negative (DN) monocytes. Whole blood was infected with 1.10^6^, 5.10^6^ or 10.10^6^ live (1, 5 and 10, respectively) and 10.10^6^ HK parasites. Cells were analyzed 4 and 24 h post-infection. (**B**) Percentages of *Li*-infected cells in each subset. (**C**) Geometric mean of CFSE parasites in CL-infected monocytes. Statistical analyses were performed using Wilcoxon test (* *p* < 0.05, ** *p* < 0.01; *n* = 7).

**Figure 4 microorganisms-10-01243-f004:**
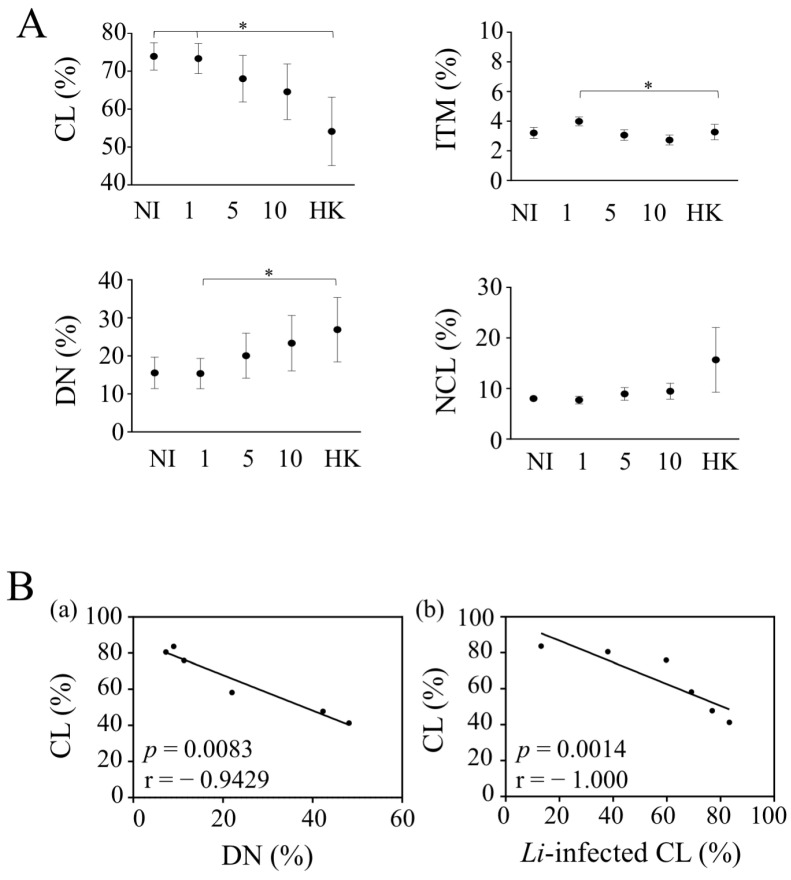
*Leishmania infantum* impacts monocyte cell subsets. (**A**) The percentages of each monocyte cell subsets were analyzed after 4 h of culture or exposition to different doses of live and heat-killed parasites (1.10^6^, 5.10^6^ and 10.10^6^/mL and HK-10.10^6^/mL). Statistical analyses were performed using a Two-Way ANOVA test with a Tukey analysis (* *p* < 0.05; *n* = 6). (**B**) (**a**) Correlation between the percentages of CL and DN. (**b**) Correlation between the percentage of CL-infected cells and the percentage of CL, 4 h after infection. Each dot represents one individual. Correlation was assessed using Spearman test, and r and *p* are shown in the figure.

**Figure 5 microorganisms-10-01243-f005:**
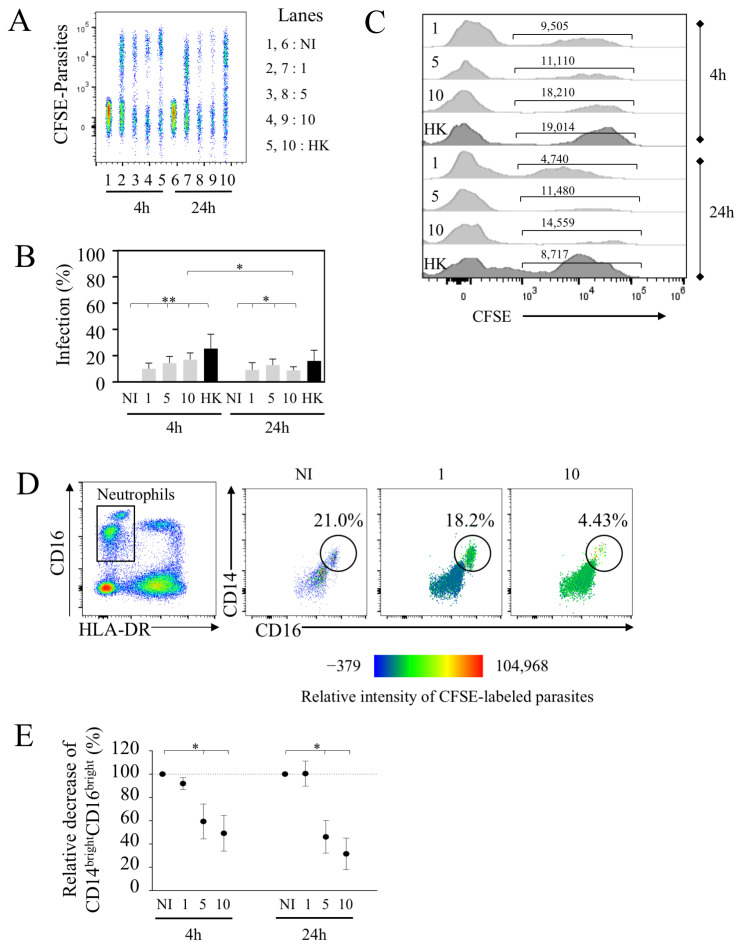
*Leishmania infantum* infects human PMNs in whole blood. (**A**) Whole blood was infected with CFSE parasites, as shown in Figure 1, using different quantities of parasites. Representative concatenation of CFSE^+^ parasites in PMNs. (**B**) Percentages of infected PMNs after 4 and 24 h of infection with different doses of parasites including live (1.10^6^, 5.10^6^ or 10.10^6^ parasites noted as 1, 5 and 10, respectively) and heat-killed (HK, 10.10^6^). (**C**) Geometric mean of CFSE parasites in infected PMNs. (**D**) Representative expression of CD14 and CD16 in PMNs and heat-map view of parasite distribution (color intensity). (**E**) Percentages of CD14^bright^CD16^bright^ PMNs are shown. Statistical analyses were performed using Wilcoxon test (* *p* < 0.05, ** *p* < 0.01; *n* = 7).

## Data Availability

All the data are included in the manuscript.

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
