# Peer review of "Leishmania infantum Infection of Primary Human Myeloid Cells"

_microorganisms, 2022, doi:10.3390/microorganisms10061243_

Round 1

Reviewer 1 Report

Τhe manuscript entitled Leishmania infantum infection of “primary human myeloid cells”, describes a no lyse no-wash assay for analyzing infection of primary human cells. The authors conclude that L. infantum promastigotes infect intermediate and classical human monocytes and they also show that a subset of PMN, CD14bright CD16bright, serves as a sanctuary for lower doses of Leishmania, as compared to CD14lowCD16+. The manuscript is well written, and it describes a straightforward protocol that can be useful for analyzing host-pathogen interactions with human cells. I think that it will be of interest to the community of researchers who investigate host-pathogen interactions

Major comments

Since this is a paper that describes a  no-wash no lyse assay method to gain insights into the infection of human monocytes with Leishmania promastigotes, I would like to see some more experimental details under the paragraph “Whole blood infection”. Ie Please describe the tubes that were used to collect the blood. Were the tubes coated with anticoagulants? In how much time after the blood donation, Leishmania parasites were added to the blood sample?

I would like to see more details on the age, the sex and the health status of individuals. And also it would be nice to be able to see in a table the different numbers of infected cells per individual (ie in a table in the supplemental files).

Did the authors measure the survival rate of parasites in different monocyte subpopulations, after several days of infection?

Minor comments

Line 14 Abstract  “(NoNo assay) Leishmania infantum”, it seems that a word is missing

In figure 1 b histogram plots labeled as Unstained, live and dead, with the histogram labeled as CFSE and identical histograms between live and dead cells, is not very clear to the reader on how the assay was performed.

Some typos like in page 5 line 198, leishmania should be Leishmania

Author Response

-I would like to see more details on the age, the sex and the health status of individuals.

For ethical issue, each donor is anonymous and this is not provided by the Établissement Français du Sang. Blood donors are between 18 and 50 years old. A sentence is included.

-And also it would be nice to be able to see in a table the different numbers of infected cells per individual (ie in a table in the supplemental files).

Although of interest this is redundant to the figures shown. Two tables could be added.

-Did the authors measure the survival rate of parasites in different monocyte subpopulations, after several days of infection?

This parameter was not performed given that the percentages of infected cells were monitored for 4 hrs and 24 hrs.

For the minor points all of them have been addressed.

Reviewer 2 Report

Picard and colleagues have designed an interesting set of experiments showing an effective flow cytometric assay capable of monitoring human myeloid cells infected with Leishmania infantum. The article is well written and very objective. It combines a relevant body of information for the audience working on intracellular infections.

I have minor comments detailed below:

L. 25 - Trypanosomatidae should be non-italicized.

L. 83 - Please, detail briefly how metacyclic parasites were obtained.

L. 96-97 - This sentence can be removed as it is explained throughout the ms.

Line 94: Please, indicate where whole blood was kept: suspension in flask, tube (vol?), agitation?

L. 123-124: Please, rewrite this sentence.

Fig.5B: Comparisons between which groups?

How viable were these parasites in infected primary monocytes? Were they capable of differentiating into amastigotes and divide within the host cell after 24h?

Author Response

All the minor points have been addressed in the revised manuscript.

Because parasite infection was assessed  after one day, this parameter (viability of parasites) was not addressed. However, as shown by confocal microscopy (Figure 2D) parasite displayed typical amastigote's morphology after overnight culture.